# Repositioning the Sm-Binding Site in *Saccharomyces cerevisiae* Telomerase RNA Reveals RNP Organizational Flexibility and Sm-Directed 3′-End Formation

**DOI:** 10.3390/ncrna6010009

**Published:** 2020-02-29

**Authors:** Evan P. Hass, David C. Zappulla

**Affiliations:** 1Department of Biology, Johns Hopkins University, Baltimore, MD 21218, USA; evan.hass@colorado.edu; 2Department of Biological Sciences, Lehigh University, Bethlehem, PA 18015, USA

**Keywords:** telomerase, Sm_7_, 3′-end formation, RNA processing, TLC1, lncRNA, senescence

## Abstract

Telomerase RNA contains a template for synthesizing telomeric DNA and has been proposed to act as a flexible scaffold for holoenzyme protein subunits in the RNP. In *Saccharomyces cerevisiae*, the telomerase RNA, TLC1, is bound by the Sm_7_ protein complex, which is required for stabilization of the predominant, non-polyadenylated (poly(A)–) TLC1 isoform. However, it remains unclear (1) whether Sm_7_ retains this function when its binding site is repositioned within TLC1, as has been shown for other TLC1-binding telomerase subunits, and (2) how Sm_7_ stabilizes poly(A)– TLC1. Here, we first show that Sm_7_ can stabilize poly(A)– TLC1 even when its binding site is repositioned via circular permutation to several different positions within TLC1, further supporting the conclusion that the telomerase holoenzyme is organizationally flexible. Next, we show that when an Sm site is inserted 5′ of its native position and the native site is mutated, Sm_7_ stabilizes shorter forms of poly(A)– TLC1 in a manner corresponding to how far upstream the new site was inserted, providing strong evidence that Sm_7_ binding to TLC1 controls where the mature poly(A)– 3′ is formed by directing a 3′-to-5′ processing mechanism. In summary, our results show that Sm_7_ and the 3′ end of yeast telomerase RNA comprise an organizationally flexible module within the telomerase RNP and provide insights into the mechanistic role of Sm_7_ in telomerase RNA biogenesis.

## 1. Introduction

Telomeres are regions of repetitive sequences at the ends of eukaryotic chromosomes that buffer against shortening caused by the end-replication problem. In most eukaryotes, telomeres are lengthened by telomerase, an RNP enzyme that adds new telomeric repeats to the ends of telomeres via reverse transcription of an RNA template [1]. Fundamentally, this process is carried out by the two core subunits of telomerase: telomerase reverse transcriptase (TERT) and the non-coding telomerase RNA [2,3]. TERT is the catalytic protein subunit of telomerase, while the telomerase RNA contains the template for reverse transcription of telomeric repeats as well as binding sites for accessory protein subunits of the telomerase RNP [4,5,6,7,8].

In *Saccharomyces cerevisiae*, the major isoform of telomerase RNA (TLC1) is 1157 nucleotides long and is predicted to form a Y-shaped secondary structure, with the template and TERT-binding regions in the central core and binding sites for holoenzyme subunits towards the tips of each arm [9,10] (Figure 1A). Like all telomerase RNAs, yeast telomerase RNAs are evolving very rapidly in sequence and size, especially in the regions between protein-binding sites [10,11]. These intervening regions can be deleted in TLC1, resulting in miniaturized “Mini-T” RNAs that still function in vivo despite being as little as one-third of the size of wild-type TLC1 [12]. It has also been shown that two important holoenzyme subunits, Est1 and Ku, retain function when their binding sites in TLC1 RNA are repositioned [10,13]. These findings show that the telomerase RNP exhibits a high degree of organizational flexibility, and they have led to the model that TLC1 acts as a “flexible scaffold”—the RNA brings together the protein subunits to form the holoenzyme but does not need to hold them in a specific position relative to one another or the catalytic core [10,13,14,15]. This flexibility also extends to human telomerase RNA, based on its rapid evolution and its tolerance to structural perturbations throughout much of its length [16,17].

In addition to the proteins Est1, Ku, and TERT (Est2), the TLC1 RNA is also bound by the Sm_7_ complex [5]. The Sm_7_ heteroheptameric protein complex is involved in biogenesis and stabilization of most spliceosomal snRNAs [22,23]. Sm_7_ binds to the consensus sequence AU_5-6_GR [24,25,26,27], which is present in TLC1 at nucleotides 1143–1150 (Figure 1B) [5]. This site is located just 7 nucleotides 5′ of position 1157, which is the 3′ end of poly(A)– TLC1 [28], the aforementioned most-abundant (“major”) isoform of TLC1. A less-abundant (“minor”) isoform, poly(A)+ TLC1, contains an extra ~100 nucleotides of TLC1 sequence on its 3′ end, as well as a poly(A) tail [21]. In addition, there are very low-abundance TLC1 transcripts terminated by the Nrd1-Nab3-Sen1 (NNS) complex that only have ~50 extra nucleotides beyond the poly(A)– TLC1 3′ end at position 1157 [19,20], but these transcripts are presumably not stable and are not detectable by northern blot.

In wild-type cells, the major, poly(A)– TLC1 isoform is present at ~29 molecules per cell, while poly(A)+ TLC1 is at only ~1 molecule per cell [29]. However, when the Sm consensus in TLC1 is mutated, only the minor poly(A)+ TLC1 isoform is detectable, likely because poly(A)– TLC1 is not stable without Sm_7_ bound [5]. Due to this critically low abundance of telomerase RNA, these mutant cells (*tlc1-Sm^–^* cells) display senescence-related growth defects (e.g., small or mixed colony sizes) but do not display a fully senescent phenotype, instead continuing to grow after the onset of these defects [5]. This “biphasic” or “near-senescent” growth phenotype (see Results Section 2.1 for a more thorough explanation of this latter term) seems to be in agreement with the fact that poly(A)+ TLC1 abundance averages only ~1 molecule per cell. Because a single molecule of TLC1 per cell is the average over a population of cells, some cells in the population probably have fewer molecules than average (i.e., none) and will eventually senesce, while others cells in the same population will have more than one molecule of TLC1 and will be able to lengthen their telomeres enough to continue dividing.

Although it was shown 20 years ago that Sm_7_ binds to telomerase RNA in yeast [5], several questions about Sm function in the telomerase RNP remain unanswered. First, while it has been shown that Sm_7_ can retain function when repositioned within Mini-T along with repositioning of the 5′ and 3′ ends of the RNA (also called “circular permutation”) [30], the flexible scaffold model has not been tested for Sm_7_ in full-length TLC1 as it has been for Est1 and Ku. Additionally, there are several open mechanistic questions regarding Sm_7_ function in TLC1 biogenesis. It was proposed that TLC1 is initially transcribed as poly(A)+ TLC1 and that this RNA is then processed into poly(A)– TLC1 [21]. It has also been proposed that the nuclear exosome exonucleolytically trims poly(A)+ TLC1 from 3′ to 5′ and is then sterically blocked at nucleotide 1157 by Sm_7_, thus generating poly(A)– TLC1 [31]. However, the hypothesis that the location of Sm_7_ binding defines the position of the mature 3′ end of poly(A)– TLC1 has remained untested, and it is still unclear whether poly(A)+ TLC1 is in fact the precursor of poly(A)– TLC1 or if NNS-terminated TLC1 transcripts are processed into the poly(A)– isoform.

Here, we show that circularly permuted full-length telomerase RNA functions in preventing senescence and, furthermore, that the Sm_7_ subunit retains its function when its binding site is repositioned in this context. This demonstrates that the Sm complex and the RNA ends are an organizationally flexible module of the telomerase RNP’s RNA scaffold. Having shown that the Sm-binding site can function at diverse positions within circularly permuted *TLC1* alleles, we next used Sm-site relocation in the context of the unpermuted RNA to test the hypothesis that the Sm-binding position defines the 3′ end of poly(A)– telomerase RNA. When we moved the Sm site to positions further 5′ in telomerase RNA, poly(A)– TLC1 RNA was still partially stabilized, and the length of the transcript was correspondingly shorter than wild type. Additionally, when we added a second Sm site 5′ of the native site in TLC1, the site at the native location still defined the mature 3′ end. This shows that binding of the Sm_7_ protein complex to telomerase RNA, in addition to providing stability, dictates formation of the mature end of the poly(A)– isoform just 3′ of its binding site, and that TLC1 processing proceeds in a 3′-to-5′ manner, likely through exonucleolytic trimming as was proposed previously.

## 2. Results

### 2.1. Sm_7_ Retains Function When Its Binding Site Is Repositioned by Circular Permutation in TLC1

To test whether the Sm_7_ protein complex retains its functions in the telomerase RNP when its binding site is repositioned in TLC1, we chose to reposition the Sm-binding site and the 3′ end together by circular permutation. Thus, Sm repositioning by circular permutation (“SmCP”) alleles allow assessment of the functions of the Sm_7_ complex at new locations in the RNP while retaining Sm-binding site location relative to the 3′ end of the RNA. We designed these *TLC1-SmCP* alleles using the TLC1 RNA secondary structure as a guide. As shown in Figure 2A, base pair position 1134 was fused to position 1 of the gene, thereby excising the endogenously encoded Sm site from its native location along with the downstream transcriptional termination sequences (Figure 1B). Then, newly encoded ends were introduced at 4 different positions in the *TLC1* sequence, while the Sm-binding region and transcriptional termination sequences (nucleotide 1130 to the end of the TLC1 locus) were appended to the new 3′ end of the gene. Additionally, to maintain endogenous expression of TLC1, the *TLC1* promoter and first 10 nucleotides from the wild-type 5′ end were retained at the beginning of the circularly permuted gene. SmCP alleles were created in this manner at positions 211, 451, and 1024 (Figure 1 and Figure 2A)—the same three positions used to reposition the Est1-binding region previously [10]. Furthermore, in order to test the positional flexibility of Sm function in all three arms of TLC1, we also created an SmCP allele in the Est1-binding arm of TLC1, at position 546.

First, to test whether these SmCP alleles maintain telomerase function and prevent senescence like wild-type TLC1, these RNAs were expressed from the *TLC1* promoter on a centromeric plasmid in a *tlc1*Δ background, and growth was monitored for 250 generations. Notably, *TLC1-SmCP@211*, *@546*, and *@1024* all supported wild-type growth throughout the 250 cell divisions (Figure 2B). This shows that the Sm_7_ complex is functioning in telomerase to allow cells to avoid senescence despite its binding site having been moved to three different locations in TLC1. In contrast to the functionality of these three alleles, tlc1-Sm^–^ and one Sm site-repositioning circular permutant, TLC1-SmCP@451, both led to a “near-senescent” phenotype in which the cells exhibited mixed or small colony sizes after 125 generations as if beginning to senesce, but then continued to grow with this colony-size phenotype through the end of passaging. While we did not observe a return to fully wild-type growth after ~100 generations as reported for *tlc1-Sm^–^* cells previously, the fact that these cells exhibit both senescence-related growth defects and long-term viability is nonetheless consistent with previous observations [5]. In an effort to accurately describe our own observations, we will continue to refer to this growth phenotype exhibited by *tlc1-Sm^–^* cells as “near-senescent” instead of the previously used descriptor of “biphasic.” To control for whether the SmCP alleles affected growth for reasons other than Sm function, we also created “Sm^–^CP” alleles, in which the repositioned Sm-binding consensus is mutated (from AUUUUUGG to UCCAACUU as in tlc1-Sm^–^) and Sm_7_-binding-incompetent [5,32]. Unlike the near-senescent cells expressing the conventional *tlc1-Sm^–^* allele, and in marked contrast to the sustained viability of cells expressing the SmCP alleles, yeast expressing any of the four Sm^–^CP alleles senesced completely by 150 generations. The long-term viability of all SmCP strains compared to the senescence of all Sm^–^CP strains strongly suggests that the Sm_7_ protein complex retains its functions at each of the four different positions to which its binding site was moved in the circularly permuted TLC1 RNAs. Furthermore, these are the first circular permutants of full-length TLC1 RNA to be tested, and it is noteworthy that the telomerase RNA ends themselves can be repositioned to any of these four locations while permitting telomerase functionality in vivo. However, because unpermuted TLC1 with a mutated Sm site (tlc1-Sm^–^) does not quite cause a senescent phenotype, the fully senescent phenotype of all of the Sm–CP alleles suggests that circular permutation of the RNA interferes with TLC1 function and/or accumulation.

Next, we assessed how TLC1 RNA processing and abundance was affected in the circularly permuted SmCP alleles. For all four SmCP telomerase RNAs, the poly(A)– isoform—predicted to be just 15 nt longer than wild type—was readily detectable by northern blotting, although at a lower abundance than wild type (Figure 2C). This shows that Sm_7_ can function in TLC1 3′-end processing and (to a lesser extent) RNA accumulation when its binding site in the RNA is repositioned. The particularly low abundance of the SmCP@451 poly(A)– RNA probably contributes to why these cells grow less well than those expressing the three other SmCP alleles (Figure 2B, lower plate). When we assessed TLC1 RNA abundance in the four Sm^–^CP alleles, we observed that poly(A)+ TLC1 was essentially undetectable, unlike in cells containing the unpermuted *tlc1-Sm^–^* allele (Figure 2C; compare lanes 9–12 with lane 4). The fact that Sm^–^CP alleles have a mutated Sm-binding site at the new positions and this results in complete loss of the poly(A)– isoform—in contrast to the SmCP alleles described above which had detectable accumulation of this RNA—provides further evidence that Sm_7_ retains partial function in promoting RNA stability when its binding site is repositioned via circular permutation. Additionally, this strongly suggests that while cells expressing tlc1-Sm^–^ accumulate just enough functional TLC1 RNA (such as the poly(A)+ isoform) to prevent complete senescence, circularly permuting tlc1-Sm^–^ reduces the already low RNA abundance to negligible levels, resulting in senescence.

We next assessed the effect of the Sm-site repositioning and circular permutation on telomere length. In the cases where cells did not senesce, we isolated genomic DNA from *TLC1-SmCP* cells at the end of passaging and subjected the DNA to Southern blotting with a telomeric probe. The results show that cells expressing *SmCP@211* and *@546* alleles had the longest telomeres, although all four SmCP constructs supported telomeres substantially shorter than wild type (Figure 2D). Telomeres were longer in *TLC1-SmCP@211* and *@546* cells than in the near-senescent *tlc1-Sm^–^* cells, providing additional evidence that Sm_7_ retains enough function when repositioned to prevent senescence. As expected from their near-senescent phenotype, *TLC1-SmCP@451* cells had the shortest telomeres of any of the SmCP alleles, averaging 212 bp shorter than wild type (lanes 7 and 8). Considering that the *SmCP@451* allele supported the shortest telomeres and the lowest RNA abundance, while the *SmCP@211* and *@546* cells had the longest telomeres and the highest RNA levels, these results suggest that relocated Sm_7_ functioned best at positions 211 and 546 and that SmCP telomerase RNAs do not support full-length telomeres primarily because of their low levels of accumulation.

### 2.2. The Sm_7_-Binding Site Defines the Mature 3′ End of Poly(A)– TLC1 RNA

It has been proposed that Sm_7_ binding to its consensus site in TLC1 (nucleotides 1143–1150) defines the mature 3′ end of poly(A)– TLC1 at nucleotide 1157 by blocking exonucleolytic trimming at the terminus of an initially longer transcript [5,31]. Although genetic data suggest that the nuclear exosome is involved in the maturation of poly(A)– TLC1 [31], the model that Sm-binding defines the mature 3′ end of poly(A)– TLC1 has not been rigorously evaluated. Having observed that Sm_7_ retains function when its binding site is repositioned in circularly permuted telomerase RNAs, we next tested the hypothesis that Sm_7_ controls 3′-end formation in poly(A)– TLC1 by moving the Sm-binding site slightly further 5′ in TLC1 without circular permutation. If the Sm-binding site’s position defines the location of the 3′ end of poly(A)– TLC1, repositioning the Sm-binding site to a position further 5′ in the RNA should result in a correspondingly truncated poly(A)– form. In the published secondary structure model of poly(A)+ TLC1 (nts 1–1251), i.e., a possible precursor to poly(A)– TLC1 that Sm_7_ could bind before RNA processing, the Sm-binding consensus is on the 5′ side of an internal loop of a hairpin, while nucleotide 1157 (the poly(A)– 3′ end) is predicted to be on the opposing side of this loop, directly across from the Sm-binding consensus (Figure 1A, inset) [10]. Additionally, the nucleotides in and around this putative internal loop show higher sequence conservation [10,30], suggesting that this predicted structure as a whole, not just the 8 nucleotide Sm-binding consensus, may be required for Sm_7_ to perform its function. Thus, based on this conservation, predicted local secondary structure, and pilot studies involving repositioning only the minimal Sm-binding consensus sequence, we chose nucleotides 1138 to 1165 as a predicted RNA module that would be likely to function when repositioned within TLC1. We then inserted this Sm-binding site into *tlc1-Sm*^–^ at positions 926, 1003, and 1089 (Figure 1A, inset). All three of these positions are predicted to be within bulged loops located on the distal portion of the RNA’s terminal arm, a region that is dispensable for telomerase function in *S. cerevisiae* [12].

TLC1 RNAs with these relocated Sm sites were expressed from centromeric plasmids in *tlc1*Δ cells, and the cells were passaged for 250 generations to test for senescence. The strains expressing tlc1-Sm^–^+Sm@1089 or @926 displayed wild-type growth, whereas those expressing tlc1-Sm^–^+Sm@1003 exhibited a near-senescent growth phenotype, similar to *tlc1-Sm^–^* cells (Figure 3A). This shows that the Sm site is functional at positions 926 or 1089 (212 and 49 nts upstream of its location in wild-type TLC1 RNA, respectively) in allowing telomerase to prevent cellular senescence. As for Sm-site function in TLC1 RNA stability and processing, northern blotting analysis revealed that, in agreement with the observed growth phenotypes, the Sm-binding sites inserted at positions 1089 and 926 in tlc1-Sm^–^ partially stabilized poly(A)– TLC1 (1% and 7% of wild-type poly(A)– TLC1 abundance, respectively), while the Sm-binding site inserted at position 1003 did not (Figure 3B). While the poly(A)– TLC1 isoform was less abundant for these 5′-shifted Sm-site alleles than it is for wild-type, the complete absence of the poly(A)– species in *tlc1-Sm^–^* cells suggests that the Sm sites at positions 1089 and 926 contribute at least some degree of function in RNA 3′-end maturation and poly(A)– RNA stability. This result is similar to what we observed with Sm site functionality in the context of circularly permuted TLC1-SmCP RNAs, which more dramatically repositioned the Sm site. Additionally, the poly(A)– RNAs that were stabilized by the inserted Sm sites at positions 1089 and 926 were shorter than wild-type poly(A)– TLC1 RNA (indicated by red arrowheads in lanes 8, 9, 12, and 13). The length of each of these poly(A)– RNAs measured from the northern blot were within 20 nucleotides (i.e., within ~2% of the RNA’s total length) of the expected size for each transcript based on the respective position of the relocated Sm site. These results provide strong evidence that the position of the Sm-binding site in TLC1 RNA defines the mature 3′ end of poly(A)– TLC1.

Sm_7_ has been proposed to define the mature 3′ end of poly(A)– TLC1 specifically by halting 3′-to-5′ exonucleolytic degradation of TLC1 [31]. As a further test of this model, we inserted the Sm site at position 1089 in TLC1 but did not mutate the native Sm site, resulting in an RNA with two different Sm sites, both with intact Sm-binding consensus sequences. If TLC1 is processed by 3′-to-5′ exonucleolytic degradation, only the native Sm site will define the mature 3′ end of poly(A)– TLC1 RNA because it is further 3′ than the Sm site at position 1089. However, if TLC1 processing occurs through a mechanism that does not proceed in a 3′-to-5′ manner (e.g., endonucleolytic cleavage), one would expect to observe a mixture of wild-type-length and truncated poly(A)– TLC1 due to processing at both the native and repositioned Sm sites. As controls, we also inserted a mutated Sm site at position 1089 in TLC1 and in tlc1-Sm^–^. When we expressed these RNAs in yeast and tested for senescence during serial passaging, we observed wild-type growth as long as the TLC1 RNA being expressed contained at least one wild-type Sm site (Figure 3C). In contrast, the *tlc1-Sm^–^+Sm^–^@1089* control exhibited a fully senescent phenotype, a more severe phenotype than the near-senescent phenotype of *tlc1-Sm^–^* cells. Northern blotting showed that the Sm site at position 1089 only directed 3′-end formation if the native Sm site was mutated (Figure 3D, lanes 12 and 13). Wild-type-length poly(A)– TLC1 was present in both *TLC1+Sm@1089* and *TLC1+Sm^–^@1089* cells (Figure 3D, lanes 8–11). These results suggest that Sm_7_ defines the mature 3′ end of poly(A)– TLC1 by halting trimming in the 3′-to-5′ direction. We also observed that inserting a Sm site at position 1089 in wild-type TLC1 reduced the abundance of poly(A)– TLC1 by 60%–70% regardless of whether or not the inserted site had an intact Sm-binding consensus sequence. Additionally, inserting a mutated Sm site at position 1089 in tlc1-Sm^–^ reduced poly(A)+ TLC1 RNA abundance below the limit of detection, which likely is the cause of the fully senescent phenotype of these cells. The results from these controls suggest that the reduced abundance of the truncated poly(A)– isoform of tlc1-Sm^–^+Sm@1089 is partially caused by inserting Sm-site sequence at 1089.

## 3. Discussion

The telomerase RNA differs in many ways from other large, well-studied, non-coding RNAs such as ribosomal and spliceosomal RNAs. Although its existence is highly conserved among eukaryotes, its sequence and length vary greatly between species [33]. Considering the rapid evolution of telomerase RNA along with experimental results has led to the model that telomerase RNA functions as a flexible scaffold for protein subunits in the telomerase RNP [10,14,15,30]. Thus, unlike other RNP enzymes that require a precise structural organization of components in the complex for function (e.g., the ribosome), the *S. cerevisiae* telomerase RNP has overall organization that has been shown to be strikingly flexible, which has provided a new paradigm for other RNP complexes, including those containing other long noncoding RNAs [10,13,14,30,34].

Here, we have shown that organizational flexibility of the large yeast telomerase RNP extends to include the Sm_7_ complex, which is a subunit of the RNP that binds just before the 3′ end of the telomerase RNA’s most abundant isoform and stabilizes it. Despite repositioning of the Sm site to four dramatically different locations across all three TLC1 RNA arms via circular permutation, Sm_7_ was still able to promote processing and stabilization of the major [poly(A)–] TLC1 isoform. Although poly(A)– TLC1 RNA abundance was reduced in these SmCP alleles (discussed below), Sm_7_ still retained partial function at all positions tested compared to negative controls. This demonstrates that Sm_7_ and its binding site in telomerase RNA function as an organizationally flexible module, along with the 5′ and 3′ ends, which were relocated along with the Sm site by circular permutation. Between the results presented here and the previously reported repositioning studies regarding the binding sites for the Est1 and Ku subunits [10,13], it is now evident that TLC1 is an organizationally flexible scaffold for these three well-established protein subunits of the telomerase holoenzyme. While organizational flexibility has not yet been reported for the Pop1/6/7 complex, which was recently reported to bind to an essential sequence in TLC1 near the Est1-binding site [8], the entire Est1-arm of TLC1 (with the Est1 site as well as the adjacent Pop1/6/7-binding site) has been shown to provide its function when expressed in trans under conditions where Est1 is artificially tethered via a heterologous binding interface to the 3′ end of TLC1 [35]. This suggests that the flexible scaffold model applies to the Pop1/6/7 proteins as well.

After determining that the Sm_7_ subunits can function in vivo when bound to different non-native positions within the telomerase RNP in the context of circular permutation, we next used Sm-site repositioning to investigate mechanistic hypotheses regarding Sm function in TLC1 RNA biogenesis. We tested whether the position of Sm binding defines the mature 3′ end of poly(A)– TLC1 by relocating the Sm-binding site further 5′ in the RNA compared to its native position. At positions where the 5′-shifted Sm site was functional in stabilizing poly(A)– TLC1, the stabilized species were correspondingly shorter RNAs, thus providing strong evidence that Sm_7_ does indeed direct 3′-end formation of poly(A)– TLC1. We also found that when wild-type Sm-binding sites were included both at the native position and at a 5′-shifted position, only the native site defined the mature 3′ end of poly(A)– TLC1, suggesting that processing proceeds in a 3′-to-5′ manner, possibly through exonucleolytic trimming, as has been previously proposed [31].

All experiments in this work were performed using *TLC1* alleles that include the endogenous promoter and terminator regions on centromeric plasmids in a *tlc1*Δ strain. Unlike the strictly single-copy expression of chromosomally integrated alleles, *CEN* plasmids are maintained from equal to a few-fold higher copy number than one per cell on average and with slightly higher variability in copy number on a cell-by-cell basis [36,37,38]. Thus, the phenotypes that we observed may be milder than if using chromosomally integrated genes, since transcript abundances may be slightly higher. However, given that our conclusions are based on qualities such as TLC1 RNA transcript length, which should not be affected by gene copy number, relative cell growth, and TLC1 abundance, in all cases compared to relevant, equally treated *CEN*-borne controls, the results reported herein still strongly support the conclusions.

The results of all of our Sm-site repositioning experiments—with or without also circularly permuting the RNA—show that although Sm_7_ did retain function when its binding site was repositioned, it did not function as well as wild type in stabilizing poly(A)– TLC1. It is possible that these defects in TLC1 RNA stabilization were caused by alterations in aspects of RNA folding. In the case of the TLC1-SmCP alleles, circular permutation may have affected local and/or global folding because of changes in the contact order of the RNA secondary structure and in the timing of when sequences emerge from RNA polymerase [39,40,41]. Altered folding of circularly permuted TLC1 RNAs may, in turn, disrupt Sm_7_ binding or make the RNA more exposed to nucleases. In addition to effects of circular permutation on TLC1 RNA abundance, misfolding could also be deleterious to telomerase activity in the case of the TLC1-SmCP@451 allele, where the Sm site and RNA ends are very close to the template-boundary element, a structure that is essential for activity in the catalytic core of the telomerase RNP [11,42]. Thus, the near-senescent phenotype of cells expressing TLC1-SmCP@451 may have been caused by decreased RNA abundance as well as impaired telomerase activity.

When we repositioned the Sm site without circular permutation, multiple RNA folding considerations may have contributed to decreased abundance of poly(A)– TLC1. First, as shown in Figure 3D, inserting the Sm site at position 1089 reduced RNA abundance even in wild-type TLC1, seemingly independent of Sm_7_ function. This may be due to disrupted folding of the region where the Sm site was inserted. Additionally, repositioning the Sm site without circular permutation may have disrupted folding of the Sm site itself. Although the portion of TLC1 that we used in the Sm-site repositioning experiments without circular permutation was chosen for its proposed hairpin structure in poly(A)+ TLC1 [10], this may not in fact be the conformation that binds, or binds best to, the Sm_7_ complex. Sm-binding sites in many other RNAs have been found to be single-stranded and preceded and/or followed by hairpin structures [24,25]. In agreement with this trend, an *Mfold* secondary structure prediction of an 1195 nucleotide NNS-terminated TLC1 RNA [19], another possible precursor of poly(A)– TLC1, shows the Sm site adopting a largely single-stranded conformation followed by a single hairpin (Appendix A). In contrast, *Mfold* secondary structure predictions of tlc1-Sm^–^+Sm@1089, @1003, and @926 all show the repositioned Sm sites forming the same hairpin-and-loop structure seen in the published secondary structure model of poly(A)+ TLC1 (Appendix A). Notably, the Sm site at position 1003, which did not stabilize poly(A)– TLC1 at all, is scored with very low *Mfold pnum* values (i.e., the region is “well determined” and predicted to have a high probability of forming the secondary structure conformation shown, as indicated by red text in Appendix A) [43], suggesting that this Sm site may have formed a stable structure that does not permit efficient Sm_7_ binding. Thus, alleles with repositioned Sm sites may not have functioned optimally because the new contexts of the Sm sites in the landscape of the TLC1 RNA did not favor folding of the most Sm_7_-binding-competent structure.

The Sm_7_ complex has a highly conserved function in the biogenesis of certain spliceosomal snRNAs, but why does it bind to *S. cerevisiae* telomerase RNA? One answer to this question may relate to an interesting theme that has emerged from the study of telomerase RNPs from a wide range of organisms. Consistent with the flexible scaffold hypothesis and the related rapid evolution of telomerase RNAs, these RNAs have independently evolved a wide variety of ways to be processed and to allow appropriate RNP biogenesis, maturation, and probably also modes of regulation [10,13,14,15,16]. Thus, the Sm_7_ complex binding to the 3′ end of TLC1 may simply be another variation on this broad theme. The Sm ring is required for 3′-end maturation of some spliceosomal snRNAs in yeast, and TLC1 has apparently “borrowed” this machinery for formation and stabilization of its own mature 3′ end [44]. However, 3′-end formation for these snRNAs seems to involve the exosome as well as other enzymes, while thus far only the exosome has been implicated in processing of TLC1 [31,45,46]. Therefore, it seems possible either that (a) these other snRNA-processing enzymes play yet-uncharacterized roles in maturation of TLC1, or (b) *S. cerevisiae* telomerase RNA has evolved a different processing mechanism from snRNAs that only requires the Sm ring and the exosome. Irrespective of these details, the recurrent observation throughout evolution that telomerase RNAs coopt proteins from other RNP complexes for their own biogenesis demonstrates both the myriad ways that noncoding RNAs can be processed and stabilized and that telomerase RNAs are exceedingly flexible in which of these mechanisms are used for their own biogenesis.

In summary, repositioning the Sm-binding site and the ends of yeast telomerase RNA has yielded insights about the organizationally flexible nature of the RNA and the mechanistic role of Sm_7_ in its biogenesis. Sm_7_ can functionally tolerate repositioning of its binding site within TLC1 along with the RNA ends. Furthermore, the telomerase enzyme’s fundamental functions can also endure the substantial reorganization of the RNA that is caused by circularly permuting the telomerase RNA transcript in vivo, despite the fact that it globally changes the RNA-folding landscape and locally breaks the phosphodiester backbone. Sm-site repositioning also has revealed that the mature 3′ end of poly(A)– TLC1 appears to be (1) controlled by the location of Sm_7_ binding to the RNA and (2) generated in a 3′-to-5′ manner, lending experimental support to previously proposed models of TLC1 biogenesis. The experimental approach of repositioning the Sm-binding site in order to change the location of the mature 3′ end could be useful for next identifying the precursor of poly(A)– TLC1 and fully characterizing telomerase RNA biogenesis in budding yeasts.

## 4. Materials and Methods

### 4.1. Construction of TLC1-SmCP Alleles

The *TLC1-SmCP* alleles were constructed using a plasmid containing two tandem copies of *TLC1* in a pRS414 backbone [47] (pDZ573, see Appendix A for full list of plasmids used). In this construct, the two copies of *TLC1* are fused into a single gene such that nucleotide 1134 of the first *TLC1* copy is followed directly by nucleotide 1 (see Figure 2A) of the second *TLC1* copy, meaning that the Sm site and transcriptional termination sequences are absent in the first *TLC1* sequence. Circular permutation was performed by PCR-amplifying specific segments of this template DNA beginning at the site of repositioning in the first *TLC1* copy and ending at the same site in the second copy (e.g., for *SmCP@211*, this resulted in a PCR product containing base pairs 211–1134 followed by 1–210). These circularly permuted DNAs were inserted into a separate plasmid containing the natural ends of the *TLC1* gene such that sequences from nucleotide 10 upstream and from nucleotide 1130 downstream (which includes the Sm site) are retained at the ends of the new gene. As an example, the sequence of *TLC1-SmCP@211* is laid out from promoter to terminator as follows: the *TLC1* promoter through nucleotide 10, 211–1134, 1–210, and 1130 through the transcriptional terminator and natural end of the *TLC1* gene (see Figure 2A). As a result of this construction scheme, nucleotides 1–10 and 1130–1134 are contained twice in all *TLC1-SmCP* alleles, making the RNAs 15 nucleotides longer than wild-type *TLC1*.

### 4.2. Experiments in Yeast

All experiments were performed in the strain TCy43 (*MATa ura3-53 lys2-801 ade2-101 trp1-1 his3-*Δ*200 leu2-*Δ*1 VR::ADE2-TEL adh4::URA3-TEL tlc1::LEU2 rad52::HIS3 pTLC1-LYS2-CEN*) [5]. All *TLC1* alleles were expressed from centromeric plasmids (Appendix A) that were derived from pSD107 (*pTLC1-TRP1-CEN*) [48]. TCy43 was transformed with *TLC1*-containing plasmids, and colonies were streaked to minimal –TRP–LYS medium. Loss of the *pTLC1-LYS2-CEN* cover plasmid was selected for by re-streaking cells to minimal –TRP medium containing α-aminoadipate. These cells were then serially re-streaked nine times to minimal –TRP medium and photographed after each round of growth. When estimating the number of generations at different points throughout passaging, each round of growth (e.g., each colony on solid medium) after loss of the cover plasmid (including the round of growth in the presence of α-aminoadipate) is approximated as 25 generations.

### 4.3. Northern Blotting

Northern blotting was performed as described previously [12,13,35,49]. Briefly, cells from the serial passaging plates were grown in liquid cultures to an OD600 of ~1.0 and harvested. Total RNA was isolated using the hot phenol method [50]. In total, 15–30 μg of total RNA from each sample was boiled, separated by urea-PAGE, and then transferred to Hybond-N^+^ Nylon membrane (GE). The membrane was UV-crosslinked and probed for both TLC1 and U1 snRNA sequence using 100-fold fewer counts of U1 probe than TLC1 probe to account for the large difference in abundance between the two RNAs. TLC1 RNA abundance was calculated by normalizing to U1 abundance, and numbers in the text and figures are expressed relative to the wild-type *TLC1* condition. Lengths of poly(A)– TLC1 RNAs in Figure 3B were calculated based on the molecular weight standard shown using ImageQuant TL (GE). The molecular weight standards used on the northern blots shown are single-stranded DNA, which migrates roughly 5% faster than single-stranded RNA. To adjust for this difference, we divided the known length of poly(A)– TLC1 (1157 nts) by its average apparent length relative to the DNA molecular weight standard to obtain an expected/observed ratio. The length measurements given by ImageQuant TL for the truncated forms of TLC1 were averaged between biological replicates and then multiplied by this ratio to produce the RNA-length measurements mentioned in the Results section.

### 4.4. Southern Blotting

Southern blotting was performed as described previously [12,13,35,49]. Briefly, cells were grown and harvested in the same manner as those used for northern blots, and genomic DNA was isolated from these pellets (Gentra Puregene system). Roughly equal amounts of genomic DNA were digested with XhoI and then separated on a 1.1% agarose gel. DNA was transferred to Hybond-N^+^ Nylon membrane (GE) to which it was UV-crosslinked. The membrane was probed for yeast telomeric sequence and a 1621 bp non-telomeric XhoI fragment that served as a non-telomeric control band [51]. Average Y′ telomere length was calculated using the weighted average mobility method described previously [13]. Briefly, ImageQuant 5.2 (GE) was used to obtain the non-Gaussian distributions of Y′ telomere signal in each lane. These distributions were aligned with one another using the mobility of their non-telomeric control bands. The weighted average mobility (WAM) of Y′ telomeric restriction fragments was calculated as follows using the pixel coordinates and intensity values (RFU) at each point across the distribution: $WAM=\frac{\sum RFU*pixel}{\sum RFU}$. The value for WAM given by this equation in pixel coordinates was then converted to base pairs using a trend line created based on a molecular weight standard on the blot. The weighted-average DNA length, in base pairs, in each lane were then set relative to the wild-type lane from their respective biological replication of the experiment, and these numbers were averaged between replicates, resulting in the values shown in Figure 2D.

## Figures and Tables

**Figure 1 ncrna-06-00009-f001:**
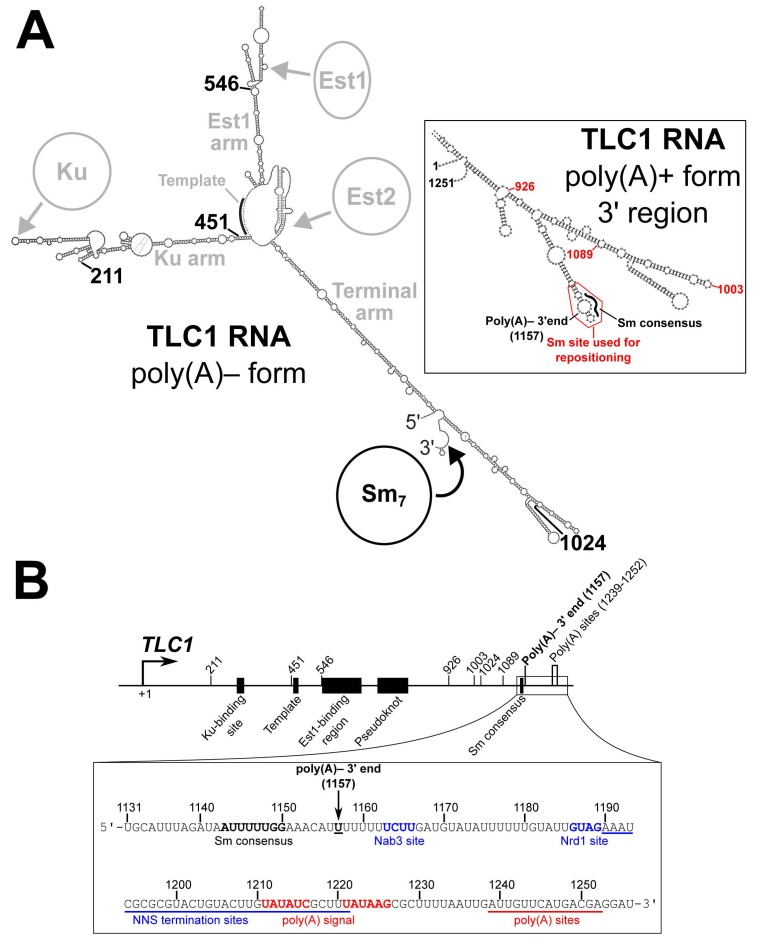
TLC1 sequence schematic and secondary structure models. (**A**) Secondary structure models of the 1157- and 1251-nucleotide (nt) forms of TLC1. The pictured secondary structure of poly(A)– TLC1 is based on previously published models [9,10,18]. The location of Sm binding is indicated in black as are the four locations to which the Sm-binding site and the 5′ and 3′ ends are repositioned in the Sm repositioning by circular permutation (SmCP) alleles used in Figure 2. The binding regions for Ku, Est1, and Est2, the template region, and the three arms of TLC1 are indicated in gray. The inset shows the secondary structure model of the 3′ region of poly(A)+ TLC1 based on a previously published model [10]. The portion of the RNA used when repositioning the Sm-binding site (nucleotides 1138 to 1165) is outlined in red. The locations of the Sm consensus and poly(A)– 3′ end are noted in black, and the three positions to which this Sm-binding site was repositioned in Figure 3 are indicated in red. (**B**) Schematic of the TLC1 gene sequence. The TLC1 regions encoding the Ku-binding site, template, Est1-binding region, pseudoknot, and Sm consensus in the RNA are denoted as black rectangles. Locations of all Sm-repositioning sites are noted by tick marks, as are the poly(A)– 3′ end and the polyadenylation sites. The inset shows the RNA sequence of the TLC1 3′ region. The Sm consensus and poly(A)– 3′ end are noted in bold. The Nab3 and Nrd1 binding sites are bolded in blue, and the region containing NNS termination sites is underlined in blue [19,20]. Similarly, the polyadenylation signal is red boldface, and the region containing polyadenylation are underscored by a red line [21].

**Figure 2 ncrna-06-00009-f002:**
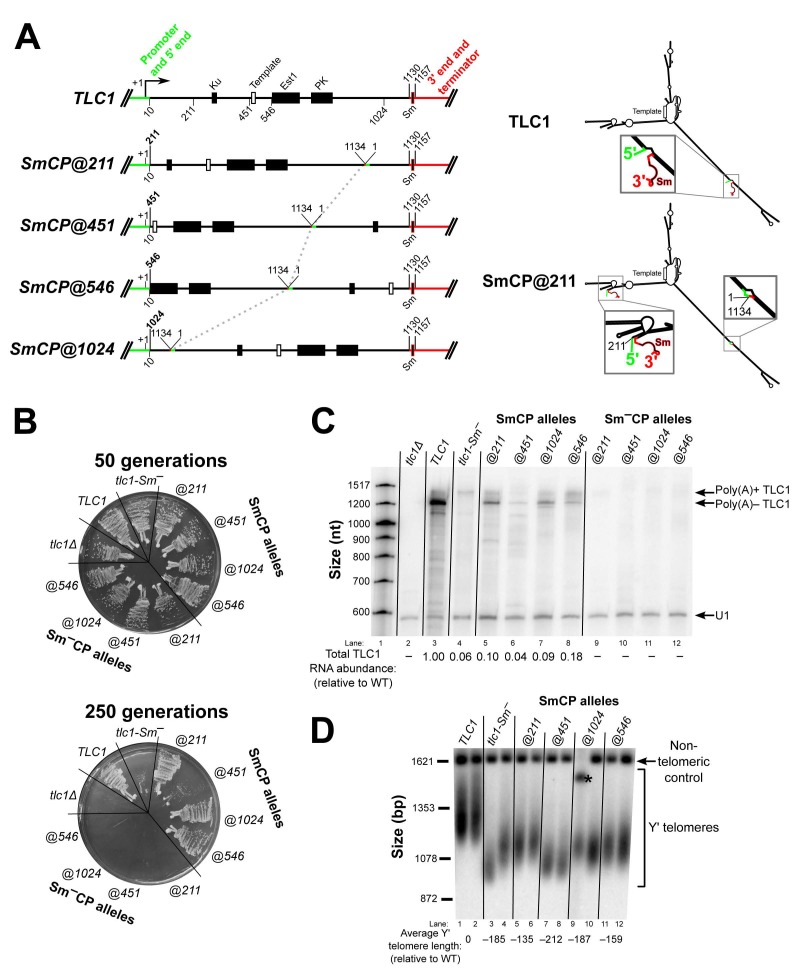
Sm_7_ retains function in telomerase RNA when its binding site is repositioned via circular permutation. (**A**) Schematics of *SmCP* gene construction and expected RNA structure. The native ends of the *TLC1* gene were effectively sealed off by connecting nucleotide 1134 with nucleotide 1, thus removing the Sm site from its native position in the RNA. A dashed line connects the location of this 1134-to-1 fusion between the 4 circularly permuted alleles. The wild-type *TLC1* promoter through nucleotide 10 (green) as well as the sequence from nucleotide 1130 on (red) flank the circularly permutated central 1–1130 region, thereby repositioning the Sm-binding site (black with red outline) along with the 5′ and 3′ ends in the encoded transcript. Black rectangles indicate the Ku-binding site, Est1-binding region, and pseudoknot (PK). A white rectangle indicates the template. The secondary structure models of wild-type TLC1 and an example of an SmCP (TLC1-SmCP@211) are schematized to the right, using the same coloring scheme as in the gene diagrams to the left. Details of the secondary structure of these large RNAs are omitted in these low-resolution schematics for the sake of simplicity, but, in fact, these regions of TLC1 have wild-type sequence and predicted secondary structure very similar to wild type as well. (**B**) *TLC1-SmCP* alleles with a wild-type Sm-binding site support sustained growth and do not cause cells to senesce. All TLC1 alleles were expressed from centromeric plasmids in a *tlc1*Δ *rad52*Δ background, and cells were serially passaged on solid media for 250 generations (10 re-streaks). (**C**) Sm_7_ confers poly(A)– telomerase RNA stabilization and appropriate 3′-end formation when repositioned via circular permutation. Total RNA was isolated from cells used in the passaging experiment shown in Figure 2B and analyzed by northern blotting with TLC1 and U1 snRNA probes. Total TLC1 RNA abundance was normalized to U1 abundance to control for sample loading in each lane and then set relative to the wild-type condition. The numbers displayed below the blot are averages of two independent biological replicates. (**D**) SmCP alleles support stable, short telomeres. Genomic DNA was isolated from cells in the passaging experiment in Figure 2B at 250 generations. Telomere length was then analyzed by Southern blotting. The blot was probed for telomeric sequence and for a 1621-bp non-telomeric restriction fragment (non-telomeric control). The pairs of lanes in the blot shown are independent biological-replicate samples, and the telomere length numbers are averages of the two replicates except in the *TLC1-SmCP@1024* condition. In this condition, the telomere length could not be quantified in the first replicate due to anomalous migration of the non-telomeric control (black asterisk), and so the displayed telomere length number in this condition is a quantitation of only the second replicate sample.

**Figure 3 ncrna-06-00009-f003:**
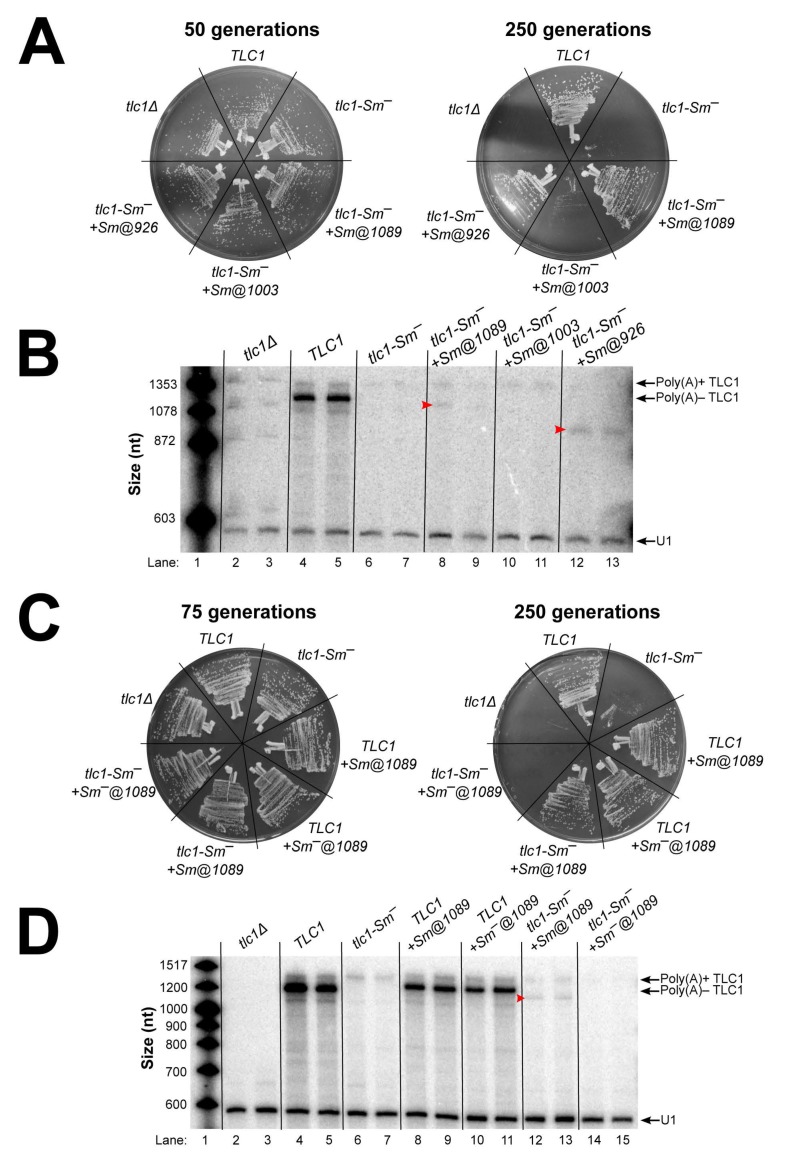
Repositioning the Sm-binding site at two of three positions further 5′ in TLC1 results in stabilization of shorter poly(A)– TLC1 RNAs. (**A**) Sm sites inserted in tlc1-Sm^–^ restored robust growth at positions 1089 and 926, but not at 1003. *TLC1* alleles were expressed and cells were passaged as in Figure 2. (**B**) Functional Sm sites inserted 5′ of the native position in tlc1-Sm^–^ stabilized shorter poly(A)– TLC1 RNAs. Total RNA was isolated from cells used in the passaging experiment in Figure 3A and analyzed by northern blotting as in Figure 2C. Red arrowheads indicate the shorter poly(A)– TLC1 RNAs stabilized in *tlc1-Sm^–^+Sm@1089* and *tlc1-Sm^–^+Sm@926* cells. The pairs of lanes represent independent biological-replicate samples. (**C**) Inserting a mutated Sm site in tlc1-Sm^–^ at position 1089 causes a fully senescent phenotype. *TLC1* alleles were expressed and cells were passaged as in Figure 2. (**D**) The native Sm site is dominant over a 5′-shifted Sm site in directing poly(A)– TLC1 3′-end formation. Total RNA was isolated from cells used in the passaging experiment in Figure 3C and analyzed by northern blotting as in Figure 2C. The truncated poly(A)– TLC1 RNA in *tlc1-Sm^–^+Sm@1089* cells is indicated with a red arrowhead. Pairs of lanes represent biological duplicate samples.

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
