# Peer review of "Repositioning the Sm-Binding Site in Saccharomyces cerevisiae Telomerase RNA Reveals RNP Organizational Flexibility and Sm-Directed 3′-End Formation"

_ncrna, 2020, doi:10.3390/ncrna6010009_

Round 1
Reviewer 1 Report
This manuscript continues to highlight the “flexible scaffold” nature of the S. cerevisiae telomerase RNA component TLC1, which controls cell replicative lifespan. In the current work, the authors focus on testing this model for the telomerase holoenzyme component, the Sm7 ring. This protein complex interacts with multiple noncoding RNAs in a structure-specific manner. In yeast telomerase, Sm7 has been proposed to confer stability to the telomerase RNA by blocking 3’ to 5’ exonucleolytic degradation, leading to a dominant RNA form with a 3’ end just downstream of the Sm7 that lacks a polyA tail. When the Sm7 binding site is mutated, this RNA species is reduced. To further define the role of Sm7 association with telomerase RNA in both biogenesis and biological function, the authors move the position of the Sm7 binding site within full-length telomerase RNA. They find moving the Sm7 ring while maintaining its adjacent position to the 3’ end generally maintains telomerase biogenesis and function, though stability is minorly affected. However, when the Sm7 binding site is moved further away from the 3’ end, this causes an alteration of the biogenesis of the TLC1 RNA that fits with the model of Sm7 blocking RNA degradation from the 3’ end. The work is clear and significantly adds to the understanding of how the Sm7 ring works with yeast telomerase.
Minor Comments
In the discussion, a brief comparison of the contribution of Sm7 to snRNA versus telomerase biogenesis would be helpful to the average reader. Pg. 1 Line 35 “…transcription of telomeric as well as…” needs correcting.Author Response
We have added a new paragraph to the Discussion section comparing the roles of Sm7 in telomerase and snRNA biogenesis.
We thank the reviewer for pointing out the error on page 1 line 35; we have corrected it in the text.
Reviewer 2 Report
Review of ncrna-703050 (Hass and Zappulla)
The yeast telomerase RNA is a long non-coding RNA that is expressed at very low levels. As TERs go, this is a rather long molecule (1157 nt) and it remains debatable why this is so. However, previous evidence strongly suggests that the RNA has distinct subdomains which do not have to occur in a strict order on the RNA; i.e. certain individual parts can be permuted on the RNA and the final molecule will still be able to assemble a functional telomerase RNP. Such permutation experiments were performed by the Zappulla group for an arm that binds to the KU proteins and one that associates with the Est1 and the Pop1/Pop6/Pop7 proteins. Hence, the idea of a modular organization of the RNP is already quite well supported.
Here then, the focus is the 3’-end of the RNA, which lacks a poly-A tail and presents a canonical Sm7 binding site just a few nucleotides upstream of the mature 3’-end. The question tackled is whether this 3’-end also can be moved to other places on the RNA, again creating circularly permuted RNAs. Furthermore, with double SM-site constructs, the question became whether the RNA is indeed formed by 3’-end processing/degradation from a slightly extended form.
For this reviewer, the second question is the more interesting one. Since there already is very good evidence that two of the three major arms can be reorganized on the RNA, it is entirely expected that the 3’-end could be permuted as well, if the structure supports a stable RNA. Furthermore, the results on question 1 (essentially Fig. 2) suffer from major drawbacks as follows:
1 - What is “near-senescence”, mentioned several times in the paper? In the original paper of the Cech lab (Seto et al 1999), they describe the growth of Sm- cells to be bi-phasic with a clear senescence phenotype at generation 100-150. How and why they recover thereafter was not investigated (did not matter for that paper). Of note, as shown by Maringele and Lydall later (Genes &Dev, 2004), even rad52D telomerase- cells can recover and grow, albeit poorly. However, this becomes an issue here and the authors should do a survivor assay, i.e. perform these experiments in RAD52+ cells. If the Sm- and the PC mutant cells become survivors in those circumstances, they were senescent, and telomerase is not sufficiently functional for keeping telomeres stable. This is particularly important for the @451 construct which seems to be very hampered and in fact in Fig. 2B looks like the tlc1D cells.
2 – How are DNA samples (Fig. 2D) derived from cells at Gen 250, if they don’t really grow (Fig. 2B; the @541 construct)? How were generations determined in these cases? These cells may be survivors, but this is not clear at all (see above).
3 - RNA preps for the Northern: there are many additional bands in most lanes; see in particular the @451 lane where some faster migrating bands are as strong as the full length one. This might mean that RNA quality is not very good, and conclusions cannot really be drawn. Or something else is happening to those RNAs in cells, but then the conclusions drawn may be flawed.
4 – As mentioned above, the @451 strain is different from the others. Why and how? How long are the telomeres in this strain for real?
5 – The analyses are complicated by the fact that this is all plasmid-based transcription. Nowadays, it is known that expression levels can vary significantly when the gene is placed on a plasmid versus at its endogenous locus in the genome. Therefore, the question arises: is the result the same when the constructs are integrated? This last point absolutely needs to be addressed with the proper experiments, otherwise the conclusions risk to be associated only to a very artificial situation and not biologically relevant.
On to the second question of how the 3’-end is generated. First, I disagree with the statement that :” It has been hypothesized that Sm7 binding to its consensus site in TLC1 (nucleotides 1143–1150) defines the mature 3’ end of poly(A)– TLC1 at nucleotide 1157 by blocking exonucleolytic trimming at the terminus of an initially longer transcript [5,31].” To me, the data in ref 5 showing that the Sm7 ring binds there is very strong. The data in Jamonnak et al (2011), combined with those in Noel et al (2012) in addition show that there are indeed such transcription stops in the predicted area, an oligo-A tail is added (the Jamonnak paper shows direct evidence for that) and resection will generate the mature poly-A- RNA. Therefore, I would not call this a hypothesis, but rather strong evidence. Now, the resection part might be investigated further, which they claim doing here. The problem is that of the three constructs tested, only two yield the expected result for 3’ to 5’ resection; one did not (the @1003 site that is actually in between the other two). I am not sure how to interpret this, but taken at face value, these results are inconsistent with a linear 3’ to 5’ resection model. Furthermore, to me there is an inconsistency at site @1089: I have difficulty understanding why the double SM- mutant is worse off than the single 3’ end SM- strain. In the start of the manuscript, it was hypothesized that the so-called ‘near-senescent’ phenotype (as opposed to the full senescent phenotype) of the SM- strain was due to poly-A+ RNA being enough to provide telomerase function. Why does this form of RNA disappear in the double SM- strain for site @1089, but not in the @1089SMwt/SM- cells (Figure 3D, lane 12, 13)? Another way to put it: the fact that @1089SM- with a 3’Sm+ is pretty much normal (Fig. 3D, lanes 10-11) says that the SM- insertion @1089 does not impinge at all on RNA transcription, folding or stability. So, again, why would the double negative be so much worse off? This is extremely confusing and must be addressed with some clearing experiments.
Round 2
Reviewer 2 Report
This revised version of the manuscript has been improved with clarifications in the text. Many of this reviewers’ initial concerns are therefore met. However, I had requested experimental clarifications of two situations: 1) the plasmid-based expression of the Tlc1 variants; 2) the @1089 situation that is difficult to understand. The authors did not add any new experimentation to the manuscript, but preferred lengthy textual explanations. This reviewer feels that particularly the “near-senescent’ phenotype could be linked to plasmid-based artifacts and must be addressed by experiments. Of course, I was not expecting that the authors repeat all experiments with integrated clean strains – I agree that would be unreasonable. However, some key experiments can easily be done in a few weeks (Sm-, and in Rad52+ as well as rad52- cells). This to me was the minimum required for this paper to go forward. Therefore, I do not feel this revision is adequate and the paper not ready for publication.
